# Characteristic Dissection of *Xanthomonas oryzae* pv. *oryzae* Responsive MicroRNAs in Rice

**DOI:** 10.3390/ijms21030785

**Published:** 2020-01-25

**Authors:** Yanfeng Jia, Chunrong Li, Quanlin Li, Pengcheng Liu, Dongfeng Liu, Zhenzhen Liu, Yanyan Wang, Guanghuai Jiang, Wenxue Zhai

**Affiliations:** 1Institute of Genetics and Developmental Biology, Chinese Academy of Sciences, Beijing 100101, China; yfjia@genetics.ac.cn (Y.J.); crli@genetics.ac.cn (C.L.); quanlinli@genetics.ac.cn (Q.L.); underway@genetics.ac.cn (P.L.); dfliu@ibcas.ac.cn (D.L.); liuzhenzhen@nibs.ac.cn (Z.L.); 15116995412@163.com (Y.W.); 2University of Chinese Academy of Sciences, Beijing 100049, China

**Keywords:** rice bacterial blight, miRNA, small RNA sequencing, degradome, regulatory units, osa-miR159b, osa-miR164a, osa-miR167d

## Abstract

MicroRNAs (miRNAs) are crucial player in plant-pathogen interaction. While the evidence has demonstrated that rice miRNAs mediate immune response to pathogens invasion, the roles of miRNAs on *Xanthomonas oryzae* pv. *oryzae* (*Xoo*) attack remain be in place. Herein, we monitored the responsive changes of rice miRNAs at 0, 8, 24 h across *Xoo* strain PXO86 infection in its compatible rice variety IR24 and incompatible variety IRBB5 by small RNA sequencing, and the genes targeted by miRNAs were also detected via degradome technology. The faithfulness of sequencing data was validated through quantitative real-time stem-loop reverse transcription-polymerase chain reaction assay. Bioinformatic analysis showed that the differentially expressed miRNAs could be divided into three immunity-related clusters, and 80 regulatory units were emerged in infection process, which comprises 29 differentially expressed known miRNAs and 38 cleaved targets. Furthermore, the miRNA presumptive function of separate immunity cluster in rice-*Xoo* interplay was confirmed through overexpressing osa-miR164a, osa-miR167d and osa-miR159b, and the disruption of regulatory units, osa-miR164a/*OsNAC60,* osa-miR167d-5p/*OsWD40-174* and osa-miR159b/*OsMYBGA*, *OsLRR-RLK2*, *OsMPK20-4*, may reset rice defense response to *Xoo* infestation in a controllable manner. These findings provide new insights into the complex roles of characteristic miRNAs and their targets in rice-*Xoo* interactions.

## 1. Introduction

Plants often answer biotic stress with growth inhibition and yield loss. Rice (*Oryza sativa* L.) supplies 25% of the total caloric intake for human, its production faces severely threat from pathogens, such as fungal, bacterial and viral invasion. Rice bacterial blight (BB), one of the most serious bacterial diseases with 20–30% yield failure, usually occurs under *Xanthomonas oryzae* pv. *oryzae* (*Xoo*) infection, and has become an ideal pathosystem to characterize the interaction between plant and bacterial pathogens [1]. However, the intrinsic defense mechanisms for BB remain relatively limited in rice. Therefore, to investigate the rice molecular regulatory mechanisms against *Xoo* infection is of great significance in understanding the rice-*Xoo* interaction and disease control.

Important advances have been made in rice response to *Xoo* attack, which lay emphasis on protein-coding genes [2,3,4,5,6,7,8,9,10,11,12], whereas the regulation events of defense response at transcriptional and post-transcriptional level remain infant. Current efforts have revealed that miRNAs are indispensable key players in rice immune system to cope with pathogens infection. In rice blast, *Magnaporthe oryzae* (*M. oryzae*), the causal fungal agent, invasion rewrites the rice immune system by inducing or suppressing the expression of numerous miRNAs. Several groups have experimentally confirmed that the transgenic rice with overexpression of osa-miR160a [13], osa-miR166k [14], osa-miR398b [13] and osa-miR7695 [15] enhance the resistance to *M. oryzae,* while the osa-miR164a [16], osa-miR167d [17], osa-miR169a [18], osa-miR319 [19] and osa-miR396 [20] negatively regulate rice immunity against the blast fungus. Likewise, comprehensive analysis of miRNAs roles in rice sheath blight has also been investigated to uncover the pathogenic mechanisms of *Rhizoctonia solani* (*R. solani*) [21], and only one paper, to date, reported that osa-miR164a promoted the rice susceptibility to *R. solani* by targeting the *OsNAC60* [16]. Several independent studies have monitored the expression alterations of miRNAs in virally-infected rice plants by small RNA profiling and/or degradome sequencing [22,23]. Experimentally, the manipulation of four miRNAs, osa-miR528 [24], osa-miR171b [25], osa-miR444 [26] and osa-miR319 [27], were involved in rice antiviral defense. In accordance with the research progress of miRNAs in rice and the aforementioned pathogens interaction, there are few published evidences describing *Xoo*-responsive miRNAs. Two recent small RNA sequencing data showed that rice miRNAs are responsive to *Xoo* utilizing dynamic and coordinated transcriptional changes and may participate in *Xa3/Xa26* mediated defense response, respectively [28,29]. Regarding applied potential, osa-miR1861k induced by multiple *Xoo* strains in resistant rice genotypes may mediate a broad spectrum resistance [30]. The inducible knockdown of osa-miR156 could motivate *OsIPA1* expression, to debar the fitness penalty of BB resistance in rice [31]. Conversely, the enhancive osa-miR169o boosted the pathogenicity of *Xoo* strains [32]. 

In this study, we examined the transcriptional changes of *Xoo*-responsive miRNAs at three time points during early infection stage and analyzed the *Xoo*-responsive genes regulated by miRNAs in the bacterial blight-susceptible variety IR24 and bacterial blight-resistant variety IRBB5, respectively. We found that 43 known miRNAs were differentially expressed (DE) in infected IRBB5 plants relative to IR24 plants, and were categorized into three immunity related groups according to their expression characteristic in the both rice varieties. Meanwhile, 71 targets negatively regulated by miRNAs were affirmed through degradome experiments. By interactive analysis on DE miRNAs and their targets, 80 miRNAs/targets regulatory units were characterized to involve in rice-*Xoo* interaction. The transgenic rice plants overexpressing miRNAs, osa-miR164a, osa-miR167d and osa-miR159b, from three independent corresponding immunity groups displayed multiple disease phenotypes as expected. Moreover, we preliminarily investigated the possible functional roles of osa-miR164a/*OsNAC60*, osa-miR167d-5p/*OsWD40-174* and osa-miR159b/*OsGAMYB, OsLRR-RLK2, OsMPK20-4* in rice defense response. 

## 2. Results

### 2.1. Diverse Bacterial Blight Response in IR24 and IRBB5 Genotypes

To dissect the potential regulatory events in transcriptional level involved in rice immunity, the bacterial blight-sensitive genotype IR24 and its near-isogenic line (NIL) IRBB5 [33], which harbor an recessive BB resistance gene, *xa5*, and enable strong defense responses to the *Xoo* strain, were chosen to challenge with *Xoo* strain PXO86 and for closer investigation. the rice genotypes at tillering stage grown in natural field were inoculated with *Xoo* strain PXO86 by leaf-clipping method, and the development of disease symptom were recorded at 14 days post inoculation (dpi) in Beijing. IRBB5 was significantly resistant (*p* < 0.01) to the *Xoo* strain PXO86, with lesion length ranging from 2.1 cm to 2.5 cm, relative to 12 cm for IR24 (Figure 1A,B). To inspect whether the transcriptional regulatory events occurred in rice-*Xoo* interactions, the expression patterns of four defense-related genes, *OsPBZ1* (Oryza sativa pathogenesis-related protein 10a), *OsPR1a* (pathogenesis-related protein 1a), *OsPR1b* (pathogenesis-related protein 1b) and *OsWRKY45*, reprogrammed by pathogen invasion, were checked by qRT-PCR at different time points post inoculation. In IRBB5 plants, the defense-responsive expressions of four genes were consistent with previous report [34,35,36,37], with markedly induced or suppressed (*p* < 0.05) at most time points, compared with those in IR24 (Figure 1C–F). All these data suggested that IRBB5 may possess different transcriptional regulatory network from IR24 in answer to BB.

### 2.2. Overview of Rice Small RNA Sequencing Data under Xoo Strain PXO86 Treatment

To validate whether small RNAs mediate the rice immunity response against the BB at a post-transcriptional level, we examined the dynamic expression change of rice small RNAs in IR24 and IRBB5 by high-throughput sequencing technology. Because of similarity of genetic background in both rice varieties, the samples gathering from IR24 were deemed to be corresponding to mock treatment. Therefore, RNA samples with three biological replicates were collected from leaves at 0, 8 and 24 h post infection (hpi) in both rice varieties, respectively. Nearly 27.8 million total reads obtained from each library after quality control were conducted to identify the potential functional small RNAs (Figure 2A). By mapping to rice reference genome, more than 87% of the clean reads had perfect matches in every sequencing data (Figure 2B). To label these mapped reads, small RNAs classification and annotation were executed according to mapping locations and alignment with corresponding small RNAs database, including miRBase, Rfam, siRNA and snoRNA. Given the focus of functional small RNAs in rice-*Xoo* interaction, we found that over 7.83% (at least 475) small RNAs sequenced from each sample were aligned to rice miRNAs (Figure 2C,E). However, the small RNAs with 24 nt in length occupied the highest abundance in all samples and 21 nt small RNAs ranked the second position (Figure 2D), suggesting numerous candidate functional siRNAs and miRNAs may exist in sequencing data. As expected, the average number of novel miRNAs and siRNAs predicted from all sample were 178 and 4912 in accordance to miRA software and biological characteristics of siRNAs, respectively (Figure 2E,F). These results implied that post-transcriptional regulatory events controlled by small functional RNAs may participate in the defense response to BB in rice.

### 2.3. The Expression Analysis of Xoo-Responsive miRNAs in Infected Rice Leaves

To uncover the receivable small RNAs functioning in *Xoo* infection progression, miRNAs detected from the sequencing data were picked to observe the dynamic change for further analysis. Small RNAs transcriptomic analysis revealed that 314 known miRNAs (accounting for 38.5% of all rice miRNAs recorded by miRBase) and 183 novel miRNAs were expressed in infected rice leaves (Appendix A). To figure out the contributor of time-point and genotype difference, principle component (PC) analysis was performed on those expressed miRNAs. PC analysis showed that PC1, explaining 25.6% of the total variance, could distinguish samples at 0 h, 8 h and 24 h, and indicated that the miRNAs-mediated regulatory network at 0 h was markedly different from 8 h and 24 h in both rice genotypes. PC2, expounding 16.2% of the total variance, could separate samples of IR24 from IRBB5, and the samples of 0 h and those at 8 h and 24 h were distributed into each sides of the plot, respectively, evidencing distinct genetic effects between IR24 and IRBB5 at all three time points (Figure 3A). These results were conformed to the immune mechanism of both rice varieties that IR24 enhances the resistance to *Xoo* with time prolongs, and the resistant *xa5* gene, restricting bacterial movement, plays a vital role in defense responses during the early stage of *Xoo* invasion in IRBB5.

*Xoo* inoculation, a biotic stimulant, undoubtedly remodeled the transcription events of the miRNAs responsible for growth and defense. Generally, the miRNAs with low expression before *Xoo* infection were sharply induced to boot protection mechanism under *Xoo* attacks where growth became defense, and vice versa. In IR24, the numbers of DE miRNAs, with a |log_2_(fold change ratio)| ≥ 1.5 expression change at examined time points, in samples of 8 h and 24 h, were 78 (36 up and 42 down) and 110 (48 up and 62 down) relative to samples of 0 h, respectively. Moreover, compared with samples of 8 h, the number in samples of 24 h was 44 (22 up and 22 down). Likewise, in IRBB5, the numbers of DE miRNAs of 8 h vs. 0 h and 24 h vs. 0 h were 135 (56 up and 79 down) and 111 (51 up and 60 down), respectively, and the number in 24 h vs. 8 h was 69 (42 up and 27 down). From time course, contrast with IR24, the number of DE miRNAs at 0 h, 8 h and 24 h were 104 (56 up and 48 down), 75 (35 up and 40 down) and 89 (43 up and 46 down) in IRBB5, respectively (Figure 3B–G, Appendix A). These data indicated that the expression level of *Xoo*-responsive miRNAs were altered inconsistently in both cultivars, and those up-regulated miRNAs, especially in IRBB5, may contribute the resistance to *Xoo* invasion in rice.

To verify the fidelity of the miRNAs expression in sequencing results, 10 known miRNAs (osa-miR1320-5p, osa-mi1432-5p, osa-miR159b, osa-miR160a-5p, osa-miR164a, osa-miR167a-5p, osa-miR167d-5p, osa-miR172a, osa-miR528-5p, osa-miR396e-5p) with divergent expression levels were selected randomly to implement qRT-PCR experiments using the rest sequencing samples and the measurement results displayed that the expression trends were basically consistent with the data of small RNA-seq (*R^2^* = 0.7164; Figure 4A), signifying that the sequenced miRNAs could be further investigated for their roles in rice immunity. To intuitively assess the impact of *Xoo* infection in both cultivars, the miRNAs expression of IR24-0 h samples served as initial value to quantify the alteration of miRNAs in others samples. By the strategy of transcripts per million (TPM) greater than 20 in at least one sample, 43 known DE miRNAs and 46 novel DE miRNAs remain creditable, and these DE miRNAs could be distributed into three classes according to their change trends upon *Xoo* attack: (I) simultaneously upregulated or downregulated in both IR24 and IRBB5; (II) elevated accumulation in IR24 but minor fluctuant or reduced accumulation in IRBB5; (III) no obvious change or repressed expression in IR24 but activated expression in IRBB5 (Figure 4B and Appendix A). Theoretically, the miRNAs of class I may contribute to basal response upon *Xoo* strain infection, while the class II and III were supposed to negatively and positively regulate the BB resistance. Subsequently, the accumulation of six miRNAs, including two basal response regulators (osa-miR1432-5p and osa-miR528-5p), two positive regulators (osa-miR159b and osa-miR167a-5p) and two negative regulators (osa-miR160a-5p and osa-miR167d-5p) chosen from the corresponding class were measured at more treatment time points by miRNA qRT-PCR (Figure 4C–H). As expected, in class I, the expression of osa-miR1432-5p was notably induced to higher levels from 0 h to 24 h after being challenged with *Xoo* strain, and osa-miR528-5p exhibited opposite alteration trends. Similarly, osa-miR159b and osa-miR167a-5p in class III and osa-miR160a-5p and osa-miR167d-5p in class II were obviously increased in IRBB5 and IR24, respectively, whereas the decreased expression or slight fluctuation were synchronously observed in the other genotype. The aforementioned results imply that miRNAs may be indeed involved in rice-*Xoo* interaction. 

### 2.4. Target Gene Identification and Function Analysis of miRNAs by Degradome Sequencing

To recognize the post-transcriptional regulatory units caused by miRNAs during rice-*Xoo* interaction, two mixed degradome libraries, IR24-D and IRBB5-D, were generated, which could provide the valuable experimental target genes validation through detecting the cleavage sites detection of miRNAs. In comparison with genomic DNA, 99.55% (34,113,226) and 99.57% (32,931,034) sequenced raw reads of IR24-D and IRBB5-D libraries were mappable, and the number of unique mappable reads were 7,267,533 and 7,068,572, respectively. More than 46% of the unique reads were mapped to the transcripts of rice protein-coding genes, which could be used for closer inspection (Appendix A). Combining with bioinformatic target prediction provided by Targetfinder software, 375 target genes were discovered for 93 miRNA families (Appendix A). 

To achieve the result accuracy of miRNAs and target genes in degradome data, the strict conditions, category ≤ 2 and raw reads ≥ 10 in at least one library, was applied to screen the target genes. Following the criteria, 71 target genes with cleaved transcripts were found in two libraries, which include 65 target genes for 31 known miRNA families and six for three novel miRNA families (Table 1). Among them, nearly 50% target genes were distributed in different transcription factors families, including *SQUAMOSA PROMOTER BINDING-LIKE* (*SPL*), *MYB*, *AUXIN RESPONSE FACTOR* (*ARF*), *NAC*, *Homeobox* (*HB*), *GRAS*, *AP2*, *GROWTH-REGULATING FACTOR* (*GRF*), *AP2/EREBP* and *MADS,* in which *NAC* and *GRF* family numbers may participate in rice immune response. For instance, osa-miR164a/*OsNAC60* [16] and osa-miR396/*OsGRFs* [20] regulatory modules manipulate rice resistance to *M. oryzae* invasion. As for target genes encoding the regulatory or metabolic enzyme, 12 targets could be divided into biotic stress related genes (*LOC_Os01g47530, LOC_Os03g02970* (*OsDCL1*), *LOC_Os03g46570, LOC_Os09g20090, LOC_Os10g41590, LOC_Os08g33370*), abiotic stress related genes (*LOC_Os08g14440, LOC_Os10g35840*) and others (*LOC_Os12g10740, LOC_Os03g55010, LOC_Os03g55030, LOC_Os04g51400*) by function annotation and sequence homology analysis. Plant hormone coordinate defense response in a synergistic or antagonistic manner [38,39]. By overlapping analysis of potential hormone and immune related to genes, seven out of 13 target genes related to biotic stress may be response to jasmonic acid (JA), auxin, gibberellin (GA), ethylene (ET), Cytokinin (CTK) salicylic acid (SA) and abscisic acid (ABA) in rice. Additionally, three genes, *OsDCL1*, *LOC_Os02g45070* (*OsAGO1a*) and *LOC_Os04g47870* (*OsAGO1b*), participated in the miRNA biosynthesis pathway and induced silencing events were also observed with differentially accumulated abundance in two libraries, indicating that miRNAs-mediated defense behavior may actually occur in rice (Table 1).

### 2.5. Multiple Xoo-Responsive miRNA/Target Regulatory Units Detected in Small RNA and Degradome Sequencing Data 

Common miRNAs, including 121 known miRNAs and nine predicted miRNAs, existed in small RNA and degradome sequencing data throughout *Xoo*-infected early stage (Appendix A). Integrating analysis on those data found that 80 miRNA/target regulatory units, encompassing 29 known miRNAs and 38 targets, and four possible miRNA/target regulatory units, including two novel miRNAs and four targets, may be responsive to *Xoo* strain PXO86 invasion in rice (Figure 5, Appendix A). Practically, in a particular miRNA/target regulatory unit, cleavage site identification of target gene is the precondition that miRNAs successfully execute their silencing function. To verify the reliability of putative *Xoo*-responsive regulatory unit, six unit members, osa-miR164a/*LOC_Os12g41680* (*OsNAC60*), osa-miR159b/*LOC_Os01g59660* (*OsGAMYB*), osa-miR1432-5p/*LOC_Os03g59770*, osa-miR167a-5p/*LOC_Os04g57610* (*OsARF8*), *LOC_Os09g39420* (*OsWD40-174*), and osa-miR167d-5p/*LOC_Os09g39420* (*OsWD40-174*) were randomly selected to show cleavage events via the degradome sequencing. In target plots (T-plots), miRNA-directed transcript cutting events were detected in corresponding position (Figure 6A–F). The inverse correlation of expression pattern between miRNA and targets was ascertained by qRT-PCR analysis, in which six miRNA/target modules displayed that all target genes were negatively regulated by corresponding miRNAs (Figure 6G–L). These results suggested that vast miRNA/target regulatory units may authentically exist in rice-*Xoo* interaction. 

### 2.6. Diversified Defense Response Caused by Osa-miR159b, Osa-miR164a and Osa-miR167d

To verify whether the *Xoo*-responsive DE miRNAs participate in rice defense response in our presumptive manner, the overexpressing (OE) rice lines of osa-miR159b, osa-miR164a and osa-miR167d, which represent the positive, basal and negative regulator of BB resistance, respectively, were generated by UBI promoter driving expression. In contrast with wild-type (WT) TP309, two independent lines with high miRNAs expression were separately identified by miRNA qRT-PCR from corresponding miRNA transgenic plants (Figure 7A–C). Then, the miRNA guided resistance in transgenic plants were examined by inoculating the *Xoo* strain PXO86. As we speculated, all osa-miR159b OE lines displayed less lesions in the leaves relative to wild type, while the lesion lengths on osa-miR167d OE and osa-miR164a OE lines were much severer than those on WT (Figure 7D,E). Consistently, strong H_2_O_2_ production was detected around the inoculation site in the leaf of osa-miR159b OE lines, and the little H_2_O_2_ was accumulated in osa-miR167d OE and osa-miR164a OE lines (Figure 7F). These results indicated that the *Xoo*-responsive miRNAs do regulate rice defense response to BB by promoting resistibility or susceptibility. Next, to ascertain whether the disease phenotype of miRNAs OE plants dependent on the form of regulatory modules, the expression level of target genes, *OsGAMYB*, *OsLRR-RLK2* (*LOC_Os12g10740*) and *OsMPK20-4* (*LOC_Os01g47530*) for osa-miR159b, *OsNAC60* for osa-miR164a and *OsWD40-174* (*LOC_Os09g39420*) for osa-miR167d-5p, were measured in corresponding miRNAs OE plants. As shown in Figure 7G–I, the transcripts of targets were significantly depressed by their reciprocal miRNA. These results suggested that different *Xoo*-responsive miRNAs may contribute to rice resistance in synergic or antergic manner, and their target genes may serve as downstream members of miRNAs to battle with BB infection. 

## 3. Discussion

miRNAs have been proven as regulator in diverse pathological progress in plants. The regulatory roles of miRNAs in bacterial blight disease process are worthy to inquiry on the excavation of new resistant genes and the deciphering of rice-*Xoo* interaction mechanism. In this study, to unveil the function of *Xoo*-responsive miRNAs at infection stage, we inspected the expression profiles of miRNAs in rice after *Xoo* treatment by small RNA sequencing. The dynamic expression changes of miRNAs were observed at three time points during *Xoo* infection, and 43 DE known miRNAs were identified in *Xoo*-infected rice leaves. Together, to uncover the target genes in miRNA-mediated defense response against BB, 71 targets with high confidence level were detected in degradome sequencing. By comprehensive analysis of those data, 29 DE known miRNAs were found to cleave 38 target genes. Furthermore, the miRNA/targets regulatory units on bacterial blight response were preliminarily validated by transgenic miRNAs overexpressing rice lines.

### 3.1. IR24 and IRBB5 Were Unbiased to Unmask the miRNAs Function in Rice-Xoo Interaction

Recent reports on miRNAs in rice-*Xoo* interaction were mostly sequencing-based, and the samples were collected from single less-resistant genotype or/and its transgenic resistant derivatives [28,29,32]. To exclude the limitation of single material and the unpredictable impact of transgenic approach and resistant gene, the rice accessions IR24 and IRBB5 were chosen to construct small RNA libraries for investigate the regulatory roles of miRNAs because of no miRNA targeting the *Xoo*-resistant gene *xa5* in bioinformatic analysis (http://plantgrn.noble.org/v1_psRNATarget/). Current evidence supports functional roles of *OsPBZ1* and *OsWRKY45* in rice-*Xoo* interaction. Rice *OsPBZ1* overexpressing plants enhanced resistance to *Xoo* infection [40], and rice allelic *OsWRKY45* genes play opposite roles in bacterial disease resistance, in which *OsWRKY45-1* and *OsWRKY45-2* are negative and positive regulator in rice resistance against *Xoo* strains, respectively [36]. By transcriptional expression examination at multiple time points, the expression of *OsPBZ1* at 8 hpi was distinct from other time points that had alike changed trends in IR24 and IRBB5; meanwhile, the *OsWRKY45-1* allelic exhibited the highest and lowest expression level at 8 hpi and 24 hpi in IRBB5, respectively, while the transcripts of *OsWRKY45-1* were gradually accumulated in 12 hpi and then decreased over time in IR24 (Figure 1F and Appendix A), suggesting the 8 hpi and 24 hpi may be ideal time points for sampling.

### 3.2. Small RNA Sequencing and Degradome Data Jointly Reveal the Post-Transcriptional Regulatory Events upon Xoo Strain Attack

An independent group has described that the expression pattern of miRNAs within the first 24 h could serve as dynamic response of rice against *Xoo* attack [28]. However, the oversimplified and less-stringent classification of miRNAs in that study may not reflect the factual responses to the *Xoo* strain, and was unfavorable to search out the functional miRNAs. Taken this into account, the DE miRNAs were designated following a more precise screen condition, miRNA in all two samples at same time-point, |log_2_(fold change ratio)| ≥ 1.5 and 20 or more reads in at least one library, and then, 43 DE known miRNAs were divided into three assumed immunity related clusters. As the gene-for-gene interaction was a conventional strategy in the battle of plant and pathogen, the target genes identification was the precondition of miRNA-mediated defense response. According to it, 375 target genes for 93 miRNA families were found through degradome sequencing. In consideration of the mixed samples in degradome experiment, we are not able to distinguish the exact contribution of each sample to occur miRNA-mediated mRNA cutting events; therefore, targets with maximum criterion were probably sliced by miRNAs, and 71 targets were affirmed. In contrast with previous works on biotic stress in rice, most targets were distributed in transcription factors [41], especially growth regulation factors, in which the majority showed the less reads in IRBB5 compared to those in IR24. One possible explanation is that the growth and development of IRBB5 genotype were inhibited by *Xoo* strain invasion, which generally causes the bigger leaves lesions. Our disease phenotype excluded the possibility of IRBB5 as such a susceptible cultivar, because the stronger resistance index was observed in IRBB5 than those in IR24 (Figure 1A). An additional explanation is the likely transcriptional reprogramming resulted from the *xa5* gene, which encodes the *γ* subunit of the basal transcription factor IIA [5]. Given that the present evidences disagreed with the first explanation, we favor the latter possibility. However, we could not monitor transcriptome changes and proteome analysis during *Xoo* strain infection, the conclusive document rests on the further investigations of them.

### 3.3. Multiple miRNA/Targets Regulatory Units Drives Flexible Defender Strategies to BB in Rice 

Recent work accumulated several lines of evidence that miRNAs/targets unit is an irreplaceable regulatory strategy in rice and pathogens interaction. To date, osa-mi169a/*OsNF-YA* [18], osa-miR166k-5p/*OsEIN2* [14], osa-miR319b-*OsTCP21* [19], osa-miR164a/*OsNAC60* [16], osa-miR396/*OsGRFs* [20] and osa-miR167d/ARF12 [17] units have been reported to participate in defense under the pressure of *M. oryzae* in rice. Moreover, osa-mi319/*OsTCP21* [27], osa-miR444/*OsMADS57* [26], osa-miR171b/*OsSCL6-IIa*, *OsSCL6-IIb*, *OsSCL6-IIc* [25] and osa-miR528/*OsAO* [24] regulatory units could change the disease development caused by rice stripe virus and rice ragged stunt virus, respectively. However, only two regulatory units, osa-miR156/*OsIPA1* [31] and osa-miR169o/*OsNF-YA* [32], were published, which mediated immunity to *Xoo*. Integrating analysis of small RNA sequencing and degradome data showed that 29 DE known miRNAs might be involved in anti-bacterial regulatory mechanism in BB. To achieve it, three miRNAs from each immunity related cluster were selected to create overexpressing plants, namely osa-miR164a OE, osa-miR167d OE and osa-miR159b OE, in the genetic background of TP309. 

Current work has proved that osa-miR164a could impair rice resistance to *M. oryzae* infection through abolishing the transcriptional activity of *OsNAC60* and this regulatory module may be conserved in plant-fungus interaction [16]. In our results, osa-miR164a represents a basal immunity regulator with reduced expression in both rice genotypes challenged with *Xoo* strain PXO86. Intriguingly, after *Xoo* infection, the expression trend of osa-miR164a in IR24 and IRBB5 was both decreased, hinting osa-miR164a may play a positive or negative role in basal defense against *Xoo*. Accordingly, our transgenetic overexpressing plants demonstrated that elevated expression of osa-miR164a, accompanying the reduced accumulation of *OsNAC60,* weakened the resistance to the *Xoo* strain (Figure 8). Therefore, as same to fungal disease in plant, osa-*miR164a/OsNAC60* module may develop similar roles in defense response to bacterial disease.

Pioneering report pointed that rice dwarf virus (RDV) and rice stripe virus (RSV) could elicit the accumulation of osa-miR167d-5p, and RSV significantly induce the expression of osa-miR167d-3p [42]. However, in this study, osa-miR167d-3p was almost no detected in all sequencing samples, and osa-miR167d-5p exhibited the noteworthy up-regulation and down-regulation in infected-samples of IR24 and IRBB5, respectively, suggesting that the inconsistent roles of osa-miR167d vary with biotic stress type in rice immunity response. Coincide with osa-miR167d mediate *M. oryzae* susceptibility [17], osa-miR167d was supposed to weaken the rice immunity response against *Xoo* strain in this study, and our experimental results have also demonstrated it through expression pattern analysis in infected rice leaves and the disease phenotype of overexpressing transgenic plants. Our data also revealed that osa-miR167d, especially osa-miR167d-5p, may act in the susceptibility regulation of *Xoo* infection in rice visa suppressing the expression of the sole target gene, *OsWD40-174,* which may maintain the silencing of *KNOX* genes throughout leaf development [43]. The reason for the defense pathway may seem that *KNOX* genes promote the lignin biosynthesis [44] (Figure 8). Although the feasible work model may be as our hypothesis, the genuine regulatory mechanism of osa-miR167d-5p mediated BB susceptibility remains elaborate in next works.

In six member of rice osa-miR159 family, osa-miR159b was the only putative Xoo resistant miRNA with the induced expression in IRBB5 and repressed transcription in IR24. The negative correlation of osa-miR159b and transcription factor OsGAMYB and protein kinase encoding genes OsLRR-RLK2 and OsMPK20-4 was experimentally validated in this study. GAMYB is an important transcriptional activator, which could function with MYBS1 to integrate nitrogen starvation and GA signaling pathways. In fact, nitrogen starvation signal and GA mutually promote the co-nuclear import of MYBS1 and MYBGA [45]. Relevant reports have shown that the high nitrogen condition may result in rice susceptibility to BB [46]. Therefore, evaluated osa-miR159b may enhance BB resistance through degrading the mRNA of *OsGAMYB*, leading to low nitrogen concentration and deficient GA signaling pathways, in rice. OsLRR-RLK2, which contains two leucine-rich repeat and a receptor-like kinase domain, was induced 24 hpi under *Xoo* infection [28], and our experimental results have also validated that osa-miR159b may increase resistance to BB visa regulating OsLRR-RLK2. Thus, OsLRR-RLK2 may be a vital player in the immune response to *Xoo* strain in rice. *OsMPK20-4* encodes a mitogen-activated protein kinase, which is induced by *M. oryzae* infection and involved in host cell death [47]. OsMPK20-4 could physically interact with OsMPK3 from a typical MAPK cascade in promoting disease resistance response [48]. We could not be sure whether independent target gene or multiple targets symbiotically participate in osa-miR159b mediated BB resistance (Figure 8); therefore, the dissection of authentic targets for osa-miR159b hangs on the following works, including genetic materials generation of the targets and their resistance evaluation.

## 4. Materials and Methods

### 4.1. Plant Materials and Growth Conditions

The *Indica* rice (*Oryza sativa* L.) cultivars, IR24 and IRBB5, were used as experimental materials to assess pathogenicity of *Xoo* strain PXO86 and construct small RNA and degradome libraries. The *Japonica* rice cultivar Taipei 309 (TP309) was served as transgenic acceptor to overexpress miRNAs. Two independent T_2_ homozygous lines with significantly increased accumulation of osa-miR159b, osa-miR164a and osa-miR167d, respectively, were applied to investigate the immune roles of miRNAs. All rice materials in this study were cultivated in the experimental field in Beijing.

### 4.2. Pathogen Inoculation and Disease Evaluation

To examine the pathological response of rice plants to BB, plants were inoculated with *Xoo* strain PXO86 at the tillering stage by the leaf-clipping method [49]. Disease index was scored by measuring the lesion length of 15 individual plants with three repeats at 2 weeks after inoculation. 

### 4.3. Small RNA Libraries Construction and Analysis

All 18 samples coming from three time points in both IR24 and IRBB5 genotypes were performed with three biological replicates for small RNA sequencing, and every sample was pooled with infected leaves collected from twenty plants in one plot to diminish the individual environment differences. Total RNA was extracted from the mentioned samples using Trizol reagent (Invitrogen, USA) according to the manufacturer’s recommendation. Small RNAs with 18~30 nt in length were isolated from the total RNA using a polyacrylamide gel electrophoresis gel, and then were sequenced by the BGISEQ-500 technology. The raw data have been uploaded into the NCBI Gene Expression Omnibus (GEO) database under the accession number GSE141995.

Raw reads were processed into clean reads by trimming adapter sequence and the low-quality reads. Then, clean reads were classified by aligning to rice genome (http://rise2.genomics.org.cn/page/rice/download.jsp) and non-coding RNA (http://rfam.janelia.org) using AASRA and cmsearch software, respectively. Reads matching mature miRNA sequence in the miRBase (v22) Database (http://mirbase.org/.) were identified as known miRNAs, and novel miRNAs were predicted using ‘miRA’ (https://github.com/mhuttner/miRA) in terms of the characteristic fold-back structure [50], and other small non-coding RNAs were also sorted out with related database. MiRNAs with at least 20 M clean reads in three biological replicates were chosen to perform expression analysis, and the expression levels were quantified by the following formula: normalized expression (transcripts per kilobase million, TPM) = mapped read count/total reads * 1,000,000. Three biological replicates were merged into mean value. The criterion of DE miRNAs was set as |log_2_(fold change ratio)| ≥ 1.5.

### 4.4. Degradome Libraries Preparation and Analysis

Two degradome libraries, IRBB5-D (including IRBB5-0 h, IRBB5-8 h and IRBB5-24 h) and IR24-D (including IR24-0 h, IR24-8 h and IR24-24 h), were generated with some modification from previously described [51]. Briefly, 5′ RNA adapters were ligated to the 5′-phosphate of degraded mRNA fragments enriched by oligo d(T) magnetic beads, then, reverse transcription were achieved with purified ligated products and biotinylated random primers. The libraries were completed via PCR-based amplification and sequenced using the Illumina HiSeq 2500. The raw data have been uploaded into the NCBI Gene Expression Omnibus (GEO) database under the accession number GSE141775.

By removing adaptor sequences and low quality reads, clean reads aligned to *Oryza sativa* transcriptome data were applied to identify potentially cleaved targets and categorized by CleaveLand pipeline [52]. The target genes corresponding to miRNAs were further deepened by overlapping the mRNA predicted by Targetfinder software. According to peak classification and value of sliced target sites, T-plot figures were built to analyze the reliability of the potential miRNA targets.

### 4.5. Vector Construction and Plant Transformation

The genomic DNA fragment harboring rice miRNA precursors amplified from TP309 were ligated into the plant binary vector UBI-pCAMBIA1300, and the overexpressing transgenic plants were created by *Agrobacterium*-mediated stable transformation in rice as previously described [53]. All primers used in the vector construction are listed in Appendix A.

### 4.6. Quantitative Reverse Transcription-PCR Analysis

Quantitative reverse transcription-PCR (qRT-PCR) was curried out to quantify the expression levels of miRNAs and their targets. MiRNAs and total RNAs were extracted from samples and reverse-transcribed to cDNA utilizing specific stem-loop reverse protocol [54] following the procedure: 16 °C for 30 min, 42 °C for 30 min, 85 °C for 5 min and ReverTra Ace qPCR RT Master Mix with gDNA Remover (Toyobo, USA) following the manufacturer’s instructions, respectively. All reactions were performed with a TransStart Green qRT-PCR Super Mix (TransGen, China) on a BioRad iQ5 sequence detection system (BIO-RAD) following the manufacturer’s instructions. The protocol used to perform qRT-PCRs was 1 min at 95 °C, followed by 40 cycles of 15 s at 95 °C and 30 s at 60 °C. U6 snRNA and *OsACTIN* gene was separately selected as the internal control for miRNAs and mRNAs, and relative expression fold changes of miRNAs and target genes were calculated by the 2^−ΔΔCt^ method with three biological replicates and at least three technical repeats. All primers used in quantitative RT-PCR analysis were listed in Appendix A.

### 4.7. 3,3′-Diaminobenzidine (DAB) Staining Assays

The physiological indexes of H_2_O_2_ accumulation in rice leaves were monitored using DAB staining, following the protocol described in a previous report [55]. In brief, the detached rice leaf sections were immersed in DAB staining solution containing 1 mg/mL DAB (Sigma). Then, the leaf samples including three biological replicates were processed by gently vacuum infiltration for 5 min and incubated at room temperature for 20 h in the dark condition. Next, the chlorophyll was removed by bleaching solution in (ethanol: acetic acid: glycerol = 3:1:1) at boiling water bath (90–95 °C) for 15 min. Finally, H_2_O_2_ accumulation can be directly visualized for the brown spots under uniform lighting. 

## 5. Conclusions

In this study, we detected 43 *Xoo*-responsive differentially expressed (DE) known miRNAs and 71 target genes in *Xoo*-sensitive rice variety IR24 and its near-isogenic line, *Xoo*-insensitive IRBB5, across *Xoo* strain PXO86 infection. By integrating analysis on DE miRNAs and their targets, 80 miRNA/target regulatory units existed in infected rice plants. We generated the transgenic overexpressing rice plants of three miRNAs, osa-miR164a, osa-miR167d and osa-miR159b, screened from independent immunity associated groups, which displayed diverse disease phenotype as expected. Meanwhile, the regulatory relationships of miRNA and targets, osa-miR164a/*OsNAC60*, osa-miR167d-5p/*OsWD40-174* and osa-miR159b/*OsGAMYB*, *OsLRR-RLK2*, *OsMPK20-4*, was also verified in the corresponding overexpressing plants. Collectively, our results revealed that multiple miRNAs/targets regulatory units are responsive to *Xoo* infection, and confirmed that the certain defensive roles of osa-miR164a, osa-miR167d-5p and osa-miR159b on *Xoo* attack.

## Figures and Tables

**Figure 1 ijms-21-00785-f001:**
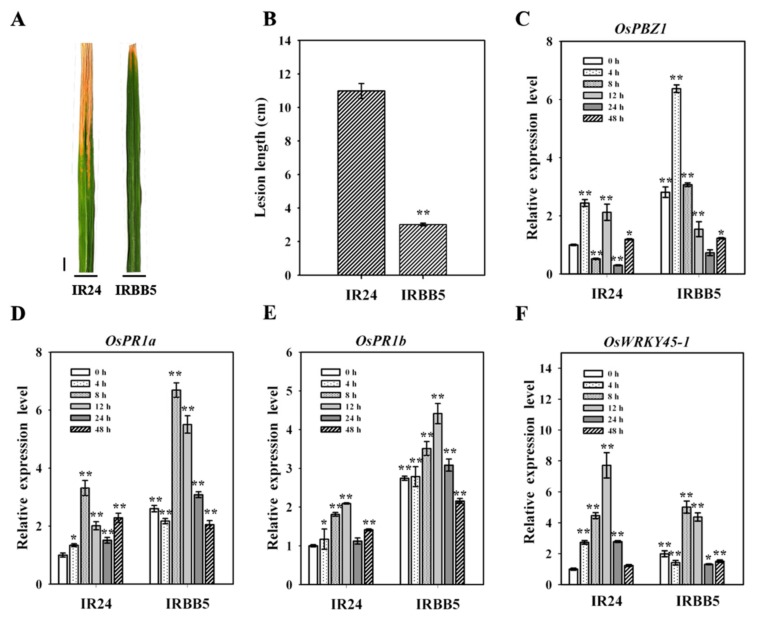
Comparison of bacterial blight responses of IRBB5 and IR24. (**A**) Disease symptom during *Xoo* inoculation of the IRBB5 and IR24. The indicated rice genotypes were inoculated with *Xoo* strain PXO86 at tillering stage in the planting field. (**B**) Lesion lengths in IRBB5 and IR24 at 14 days post inoculation with *Xoo* strain PXO86. (**C**–**F**) Expression pattern of four published defense-responsive genes in IRBB5 and IR24 under *Xoo* strain PXO86 treatment. RNA was extracted at the indicated time points for qRT-PCR experiments, and the relative expression values of all samples were normalized by quantifying the miRNAs transcription of IR24 0 h samples. Scale bar, 1 cm. Bars represent mean ± standard deviation (*n* = 3 independent biological replicates). Asterisks indicate a statistically significant difference (** *p* < 0.01, * *p* < 0.05).

**Figure 2 ijms-21-00785-f002:**
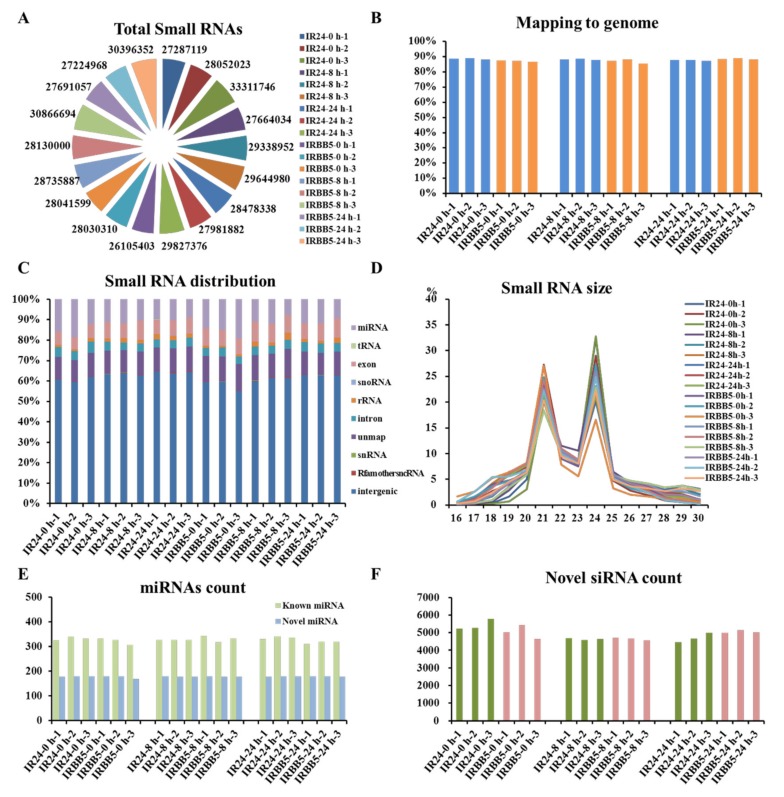
Profiling of small RNA sequencing in two rice genotypes inoculated with *Xoo* strain PXO86. (**A**) Total clean reads distribution in libraries from different time point treatment with *Xoo* strain PXO86. (**B**) The mapping percentage of the clean reads in 18 sequenced samples. (**C**) The classification distribution of the sequenced small RNA for every rice small RNA library. (**D**) Length distribution and abundance of the small RNAs in each library. (**E**) The statistic of known and candidate miRNAs in disparate sample. (**F**) The distribution number of predicted siRNAs in disparate sample. ‘IR24′ represents the rice cultivar ‘IR24′; ‘IRBB5′ represents the rice cultivar ‘IRBB5′; ‘0 h’, ‘8 h’ and ‘24 h’ represents separately the treatment time point with *Xoo* strain PXO86; ‘−1′, ‘−2′ and ‘−3′ means the different biological replicate, respectively; nt, nucleotide.

**Figure 3 ijms-21-00785-f003:**
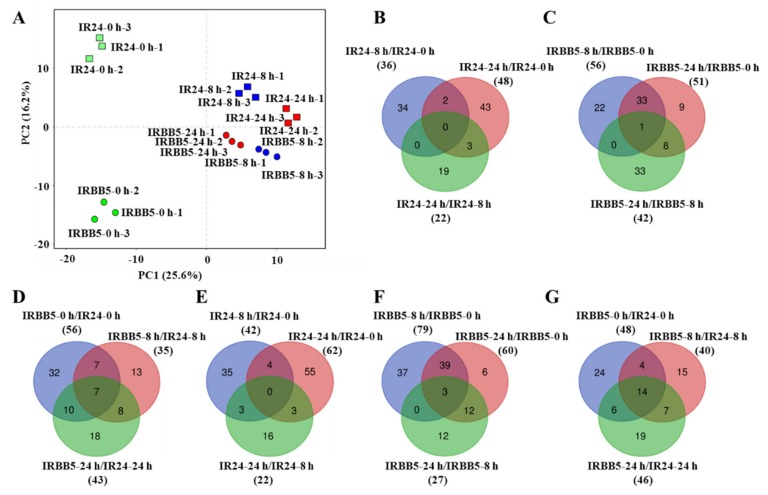
The overview of the miRNAs changes in IR24 and IRBB5. (**A**) Principle component analysis of detected miRNAs in small RNA sequencing data. Green, blue and red box denotes samples at 0, 8 and 24 h after treatment with *Xoo* strain PXO86 in IR24, respectively. Similarly, green, blue and red dot denotes samples at 0, 8 and 24 h after treatment with *Xoo* strain PXO86 in IRBB5, respectively. The samples at different time point have three biological replicates. (**B**–**D**) Venn diagrams of up-regulated differentially expressed miRNAs (DE miRNAs) in IRBB5 and/or IR24 after PXO86 strain infection. (**E**–**G**) Venn diagrams of down-regulated DE miRNAs in IRBB5 and/or IR24 after PXO86 strain infection.

**Figure 4 ijms-21-00785-f004:**
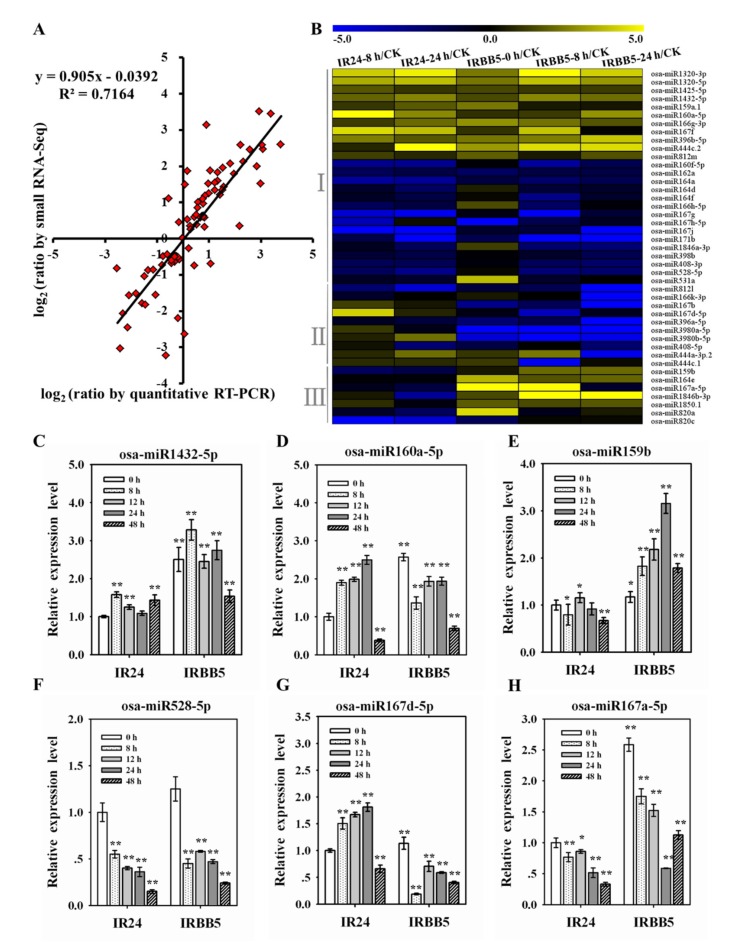
The responsive expression changes of miRNAs after *Xoo* infection. (**A**) The correlation analysis of miRNA expression level between small RNA sequencing and qRT-PCR experiments. (**B**) Heat map of the documented miRNAs with |log_2_ (fold change ratio)| ≥ 1.5 across *Xoo* infection. Yellow and blue colors represent up-regulated and down-regulated expression level, respectively. CK represents IR24-0 h. (**C** and **F**) The expression profile of basal immunity regulators, osa-miR1432-5p and osa-miR528-5p, in response to the *Xoo* strain. (**D** and **G**) The expression profile of negative immunity regulators, osa-miR160a-5p and osa-miR167d-5p, in response to the *Xoo* strain. (**E** and **H**) The expression profile of positive immunity regulators, osa-miR159b and osa-miR167a-5p, in response to the *Xoo* strain. The miRNA expression levels in all samples were quantified relative to IR24-0 h. Bars represent mean ± standard deviation (*n* = 3 independent biological replicates). Asterisks indicate a statistically significant difference (student *t*-test, ** *p* < 0.01, * *p* < 0.05).

**Figure 5 ijms-21-00785-f005:**
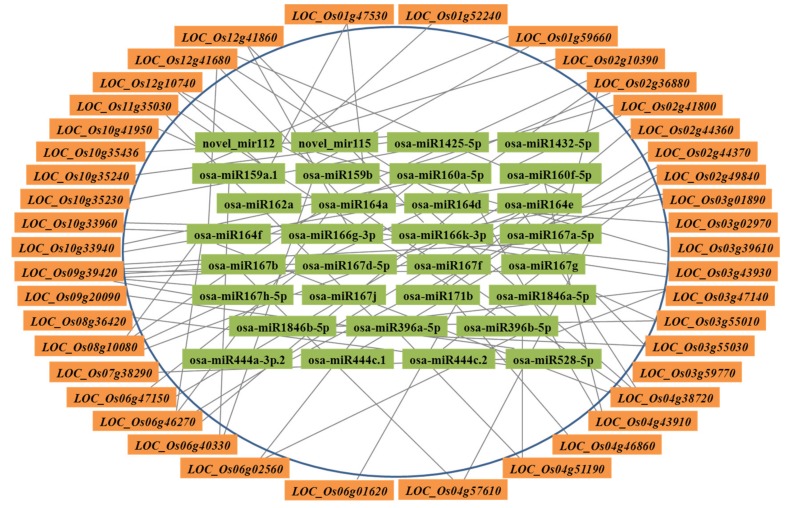
The *Xoo*-responsive regulatory units of miRNA/targets in infected rice plants. Numbers in internal block with light green color were miRNAs. The genes in outer block represent the targets for the counterparts in the line.

**Figure 6 ijms-21-00785-f006:**
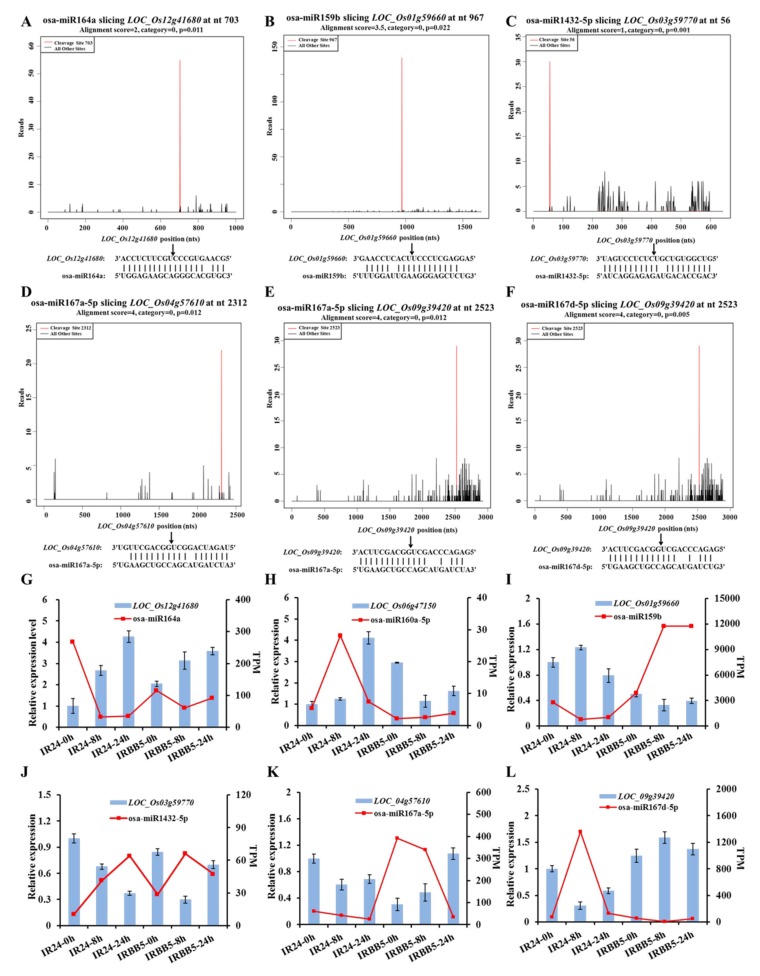
The negative correlation identification of miRNA and target genes. (**A**–**F**) Target plots (T-plots) of representative miRNA targets validated by degradome sequencing. The red lines showed the distribution of the degradome tags along the target mRNA sequences. The black arrows represented the cleavage sites of miRNAs on the target genes. (**G**–**L**) The relative expression levels of miRNA and targets at different time points in *Xoo*-infected IRBB5 and IR24. The histogram and red dot indicate the expression levels of targets from quantitative reverse transcriptase-polymerase chain reaction (qRT-PCR) analysis and miRNA sequencing data, respectively.

**Figure 7 ijms-21-00785-f007:**
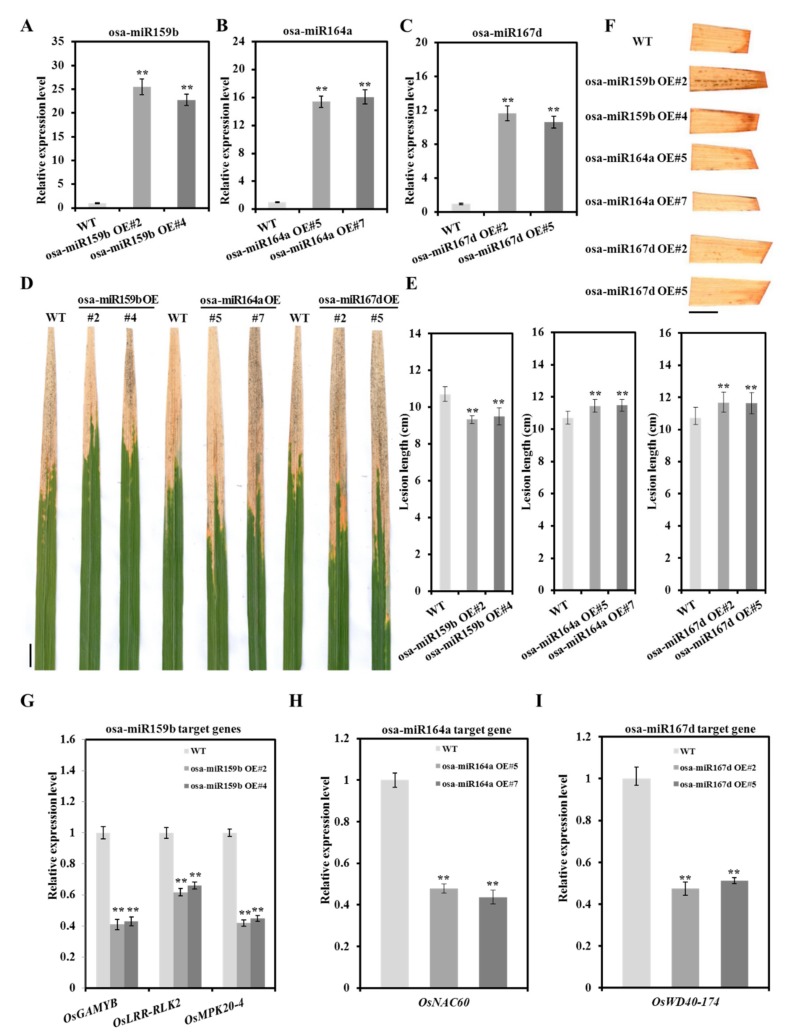
Multiple disease phenotype mediated by osa-miR159b, osa-miR164a and osa-miR167d-5p in overexpressing transgenic plants. (**A**–**C**) The excessive accumulation of osa-miR159b, osa-miR164a and osa-miR167d-5p in overexpressing transgenic lines. The wild-type (WT) Taipei 309 was used as control. (**D** and **E**) Diverse disease phenotypes caused by the *Xoo* strain PXO86 in different overexpressing transgenic lines. Lesion lengths for 45 diseased leaves in 15 plants were statistically analyzed at 14 dpi. (**F**) The histochemical detection of H_2_O_2_ using 3,3′-diaminobenzidine (DAB) staining in WT and overexpressing transgenic lines after *Xoo* infection. Scale bar, 1 cm. (**G**–**I**) The relative expression levels of targets in overexpressing transgenic lines. Bars represent mean ± standard deviation (*n* = 3 independent biological replicates). Asterisks indicate a statistically significant difference (student *t*-test, ** *p* < 0.01, * *p* < 0.05).

**Figure 8 ijms-21-00785-f008:**
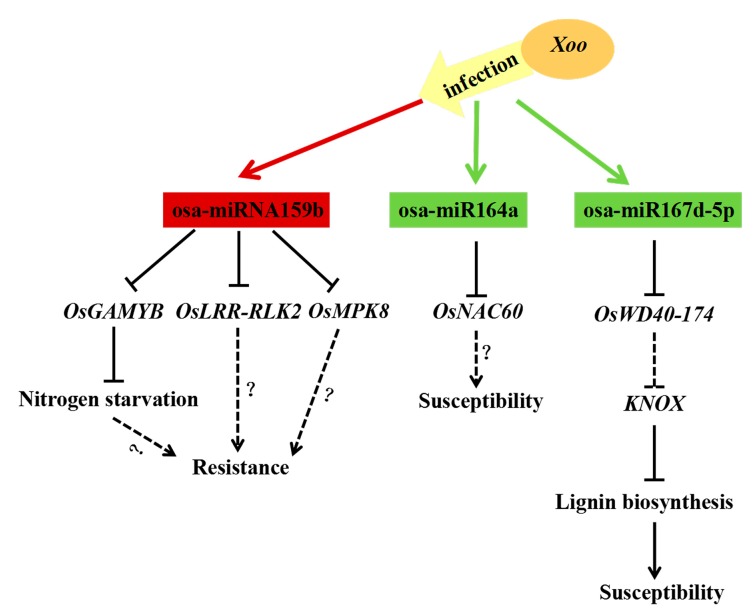
The potential work model of osa-miR159b, osa-miR164a and osa-miR167d-5p mediated immunity pathways upon *Xoo* infection. The up-regulated and down-regulated miRNAs are shown in red and green, respectively. Solid lines and dashed lines separately represent confirmative and putative regulatory links in rice or *Arabidopsis*.

**Table 1 ijms-21-00785-t001:** miRNA Targets assembly with category ≤ 2 and raw reads ≥ 10 in at least one library.

miRNA	Target	Cleavage Site (nt)	Category	Target Annotation	Response Type	Raw Reads in IRBB5	Raw Reads in IR24
osa-miR1425.	*LOC_Os10g35436.*	1213	0	Pentatricopeptide repeat protein 1	Bacterium, fungus, innate immunity/ Jasmonic acid (JA)	34	92
	*LOC_Os10g35230*	1192	0	Pentatricopeptide repeat protein 2		10	13
	*LOC_Os10g35240*	1219	0	Pentatricopeptide repeat protein 4		42	57
osa-miR1426	*LOC_Os10g06130*	829	0	RNA recognition motif containing protein	Jasmonic acid (JA)	13	12
osa-miR1428	*LOC_Os03g17980*	1228	0	Serine/threonine protein kinase		2	10
osa-miR1432	*LOC_Os03g59770*	56	0	calcium-binding protein CML21-like		30	48
osa-miR1437	*LOC_Os01g22490*	381	0	Ubiquitin 5		13	9
osa-miR156	*LOC_Os06g45310*	759	0	Squamosa promoter binding-like 11		10	30
	*LOC_Os02g04680*	1158	0	Squamosa promoter binding-like 3		21	34
	*LOC_Os08g41940*	1008	0	Squamosa promoter binding-like 16		9	7
	*LOC_Os06g49010*	1155	0	Squamosa promoter binding-like 12		17	11
osa-miR159	*LOC_Os01g59660*	967	0	GAMYB-like protein 1	Gibberellin (GA)	140	280
	*LOC_Os12g10740*	1018	0	Leucine-rich repeat family protein		2	10
	*LOC_Os01g47530*	1664	0	Mitogen-activated protein kinase 20-4	Basal defense	7	31
	*LOC_Os06g40330*	1429	0	MYB transcription factor		12	25
osa-miR160	*LOC_Os02g41800*	1340	0	Auxin response factor 8		134	105
	*LOC_Os10g33940*	1367	0	Auxin response factor 16	Auxin	1212	1347
	*LOC_Os06g47150*	1379	0	Auxin response factor 18	Auxin	2407	2987
	*LOC_Os04g43910*	1355	0	Auxin response factor 10		48	59
osa-miR162	*LOC_Os03g02970*	2988	0	Dicer-like protein 1		24	42
osa-miR164	*LOC_Os08g10080*	697	2	NAC transcription factor 104		17	2
	*LOC_Os04g38720*	697	0	NAC transcription factor 2	Gibberellin (GA)	29	7
	*LOC_Os12g41680*	703	0	NAC transcription factor 60	*Magnaporthe oryzae*	55	140
	*LOC_Os06g46270*	679	0	NAC transcription factor 11	Auxin	28	20
	*LOC_Os02g36880*	787	0	NAC transcription factor 1		72	47
osa-miR166	*LOC_Os03g43930*	622	0	Homeobox-leucine zipper protein HOX32	Auxin	66	143
	*LOC_Os12g41860*	613	0	Homeobox-leucine zipper protein HOX33	Auxin	66	142
	*LOC_Os03g01890*	574	0	Homeobox-leucine zipper protein HOX10	Auxin	266	277
	*LOC_Os10g33960*	580	0	Homeobox-leucine zipper protein HOX9	Auxin	267	278
osa-miR167	*LOC_Os04g57610*	2312	0	Auxin response factor 8	Auxin	22	8
	*LOC_Os09g39420*	2523	2	WD40 repeat-like domain containing protein		3	29
osa-miR168	*LOC_Os02g45070*	427	2	Protein argonaute 1A	Virus, innate immunity/Auxin	31	34
	*LOC_Os04g47870*	535	2	Protein argonaute 1B	Virus, innate immunity /Auxin	6	19
osa-miR171	*LOC_Os06g01620*	452	0	GRAS plant-specific transcription factor		354	367
	*LOC_Os04g46860*	1130	0	GRAS plant-specific transcription factor		206	328
	*LOC_Os02g44370*	1139	0	GRAS plant-specific transcription factor		80	75
	*LOC_Os02g44360*	1121	0	GRAS plant-specific transcription factor		73	73
osa-miR172	*LOC_Os05g03040*	1423	2	AP2/EREBP transcription factor	Ethylene (ETH)	1728	1728
	*LOC_Os07g13170*	1207	0	AP2 transcription factor		21	43
osa-miR1846	*LOC_Os03g55010*	1219	0	UDP-glucosyltransferase family protein		6	10
	*LOC_Os03g55030*	1213	0	UDP-glucosyltransferase family protein		6	10
osa-miR2093	*LOC_Os09g06499*	1549	2	Sulfate transporter 4.1		18	10
osa-miR2094	*LOC_Os08g14440*	506	2	Uridylyltransferase-related	Cytokinin (CTK)	26	14
osa-miR2100	*LOC_Os05g27940*	250	2	Ribosomal protein S7		7	11
osa-miR2102	*LOC_Os03g46570*	902	2	SKIPa-interacting protein 31	Bacterium, fungus/ Gibberellin (GA)	11	3
	*LOC_Os09g34140*	183	2	Conserved hypothetical protein		17	12
osa-miR2925	*LOC_Os07g25430*	235	2	Photosystem I reaction center subunit IV, chloroplast precursor	Cytokinin (CTK)	42	21
	*LOC_Os01g45470*	82	2	Conserved hypothetical protein		32	34
osa-miR2926	*LOC_Os10g35840*	1175	2	Glutamyl-tRNA reductase 2		31	22
osa-miR393	*LOC_Os05g05800*	1552	2	F-Box auxin receptor protein	Defense response/ Auxin, Ethylene (ETH)	66	102
	*LOC_Os04g32460*	1513	2	Auxin signaling F-box 2	Defense response/Auxin	23	20
osa-miR394	*LOC_Os01g69940*	1093	2	Cyclin-like F-box domain containing protein	Auxin	165	169
osa-miR396	*LOC_Os06g02560*	410	2	Growth-regulating factor 5		62	55
	*LOC_Os11g35030*	593	0	Growth regulating factor protein 8	Gibberellin (GA)	31	27
	*LOC_Os04g51190*	461	2	Growth-regulating factor 3	Gibberellin (GA)	18	13
	*LOC_Os03g47140*	572	2	Growth regulating factor protein 9		70	105
osa-miR399	*LOC_Os04g55230*	3898	2	Tetratricopeptide repeat protein	Abscisic Acid (ABA)	7	22
	*LOC_Os09g39180*	731	2	Similar to Nucleic acid-binding protein precursor	Innate immunity Abscisic Acid (ABA)	10	17
osa-miR414	*LOC_Os08g33370*	725	2	G-box factor 14-3-3 protein	*Xoo*, *Magnaporthe oryzae*/ Jasmonic acid (JA), Ethylene (ETH), Abscisic Acid (ABA)	1	11
osa-miR444	*LOC_Os02g49840*	290	2	MADS-box transcription factor 57		147	211
osa-miR528	*LOC_Os09g20090*	32	2	Ascorbate oxidase	Antiviral defence/ Salicylic Acid (SA)	3	23
	*LOC_Os07g38290*	538	2	Cupredoxin domain containing protein		837	1780
	*LOC_Os08g36420*	80	2	Zinc/iron permease family protein		8	14
osa-miR530-5p	*LOC_Os04g51400*	510	2	RING finger protein with microtubule-targeting domain 1		605	1107
osa-miR5809	*LOC_Os12g25180*	403	2	Hypothetical conserved gene		29	35
novel_mir112	*LOC_Os10g41590*	1206	2	Flowering-Related RING Protein 1	Innate immunity	5	16
novel_mir89	*LOC_Os02g04680*	1156	0	Squamosa promoter binding-like 3		77	36
	*LOC_Os06g45310*	760	0	Squamosa promoter binding-like 11		14	41
novel_mir115	*LOC_Os03g39610*	743	0	chlorophyll A-B binding protein	Reactive oxygen species Biosynthesis/ Abscisic Acid (ABA)	85	70
	*LOC_Os02g10390*	706	0	chlorophyll A-B binding protein	Cytokinin (CTK)	61	59
	*LOC_Os01g52240*	749	2	chlorophyll A-B binding protein		3158	4197

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
