# Peer review of "Characteristic Dissection of Xanthomonas oryzae pv. oryzae Responsive MicroRNAs in Rice"

_ijms, 2020, doi:10.3390/ijms21030785_

Round 1
Reviewer 1 Report
The authors present the work: "Characteristic dissection of Xanthomonas oryzae pv. oryzae responsive microRNAs in rice "
It is a quite extensive work including miRNA sequencing and detection, RT-qPCR , degradome analysis, transgenic plants construction and challenging with Xoo, etc.
My main suggestion would be to include some data they may already have. In the degradome analysis they show data for osa-miR159b, osa-miR164a and osa-miR167a target genes (OsGAMYB, OsNAC60 and OsARF8, respectively) and later on they show their results for transgenic plants overexpressing osa-miR159b, osa-miR164a and osa-miR167d. So, I find that even if the say that degradome units were randomly selected, and them they make constructs and transgenic plants which are altered for the expression of those miRNAS, except for osa-miR167a, instead they have a transgenic line for osa-miR167d. I would therefore include degradome analysis for osa-miR167d target gene OsWD40-174.
Secondly when they are analysing results for lines IR24 and IRBB5 it is not clear which comparisons are made when they show **: significant differences (I have marked this in the text)
And finally I suggest that along all the text they should use the same nomenclature for miRNAs: osa-miR159 instead of miR159.
There are also some spelling mistakes, marked in the text, and some sentences that need revision, also marked in the text.
Author Response
1. My main suggestion would be to include some data they may already have. In the degradome analysis they show data for osa-miR159b, osa-miR164a and osa-miR167a target genes (OsGAMYB, OsNAC60 and OsARF8, respectively) and later on they show their results for transgenic plants overexpressing osa-miR159b, osa-miR164a and osa-miR167d. So, I find that even if the say that degradome units were randomly selected, and them they make constructs and transgenic plants which are altered for the expression of those miRNAS, except for osa-miR167a, instead they have a transgenic line for osa-miR167d. I would therefore include degradome analysis for osa-miR167d target gene OsWD40-174.
Reply: Thanks for your positive comment, and we have add degradome analysis for osa-miR167d target gene OsWD40-174 and osa-miR167a target gene OsWD40-174 in the updated manuscript (please see line 271-272, Figure 6E-F, 6L).
2. Secondly when they are analysing results for lines IR24 and IRBB5 it is not clear which comparisons are made when they show **: significant differences (I have marked this in the text)
Reply: Thanks for your positive comment, and we have revised it in the updated manuscript (please see line 104-106, line 190-192 and line 221).
3. And finally I suggest that along all the text they should use the same nomenclature for miRNAs: osa-miR159 instead of miR159.
Reply: Thanks for your positive comment, and we have revised it in the updated manuscript (please see line 21-23, 27, 46-48, 51, 55, 61, 62, 64, 74, 77, etc and Figure 4, Figure 5, Figure 6, Figure 7 and Figure 8).
4. There are also some spelling mistakes, marked in the text, and some sentences that need revision, also marked in the text.
Reply: Thanks for your positive comment, and we have revised it in the updated manuscript (please see line 10, line 12, line 19, line 31, line 44, line 56, line 67, line 87, line 124, line 261, line 368, line 374, line 386, line 461, line 474-476, line 491-498).
Reviewer 2 Report
Involvement of small RNAs in the immunity response against pathogen including Xoo is currently recognized. This manuscript characterizes microRNAs in rice in relation to Xoo infection. The aim of this work is undoubtedly important for researchers in the related filed, but the data presentation and the interpretation of the provided data in this manuscript are premature.
First, the authors use two varieties, IR24 and IRBB5 for a comparison of miRNA abundance. From the results, for example Fig. 3A and 3D, it is evident that more than 100 miRNAs show different abundances between IR24 and IRBB5 before Xoo inoculation. However, it is not clear how the differences before Xoo inoculation is associated with resistance seen in IRBB5. The authors should answer and analyze how the differences before Xoo inoculation contributes immune responses. In addition, the authors need to clarify how changes in miRNA abundance after Xoo infection are related to the difference in resistance.
Second, are time points indeed sufficient to characterize microRNA dynamics in the response to Xoo infection? In Fig. 1c, OsPBZ1 expression is obviously induced 4h after inoculation, then its expression is reduced 8h after inoculation. This result suggests that transcriptome changes are so dynamic from 4h to 8h after infection. Is it not necessary to test miRNA expressions during that time?
In addition to the above comments, I would like to raise several points that the authors should revise in order to make the arguments in the manuscript clearer.
Line 125-127, the data provided in this paragraph are the result of small RNA sequencing. The result do not have any relationship with defense response. Line 151-154, I can not agree that these results were confirmed to …. In Figure 3B-D, it is better to compare between up-regulated miRNAs and to compare down-regulated miRNAs, separately. Line 185-191. miR167a-5p and miR167d-5p do not regulate common targets? This point is not clear in the manuscript. In addition, the phrases “positive regulators” and “negative regulators” are used for describing miR167a-5p as a positive and miR167d-5p as a negative. These phrases confuse me. Please reconsider these phrases. Samples used in figure 4A are not described. Figure 4B must use colors with careful consideration of color weakness. Figure 4C-H, a statistically significant deference compare to what? Figure 5 seems not to easy to understand the authors argument. Rewrite the figure.
Author Response
Review2
1. First, the authors use two varieties, IR24 and IRBB5 for a comparison of miRNA abundance. From the results, for example Fig. 3A and 3D, it is evident that more than 100 miRNAs show different abundances between IR24 and IRBB5 before Xoo inoculation. However, it is not clear how the differences before Xoo inoculation is associated with resistance seen in IRBB5. The authors should answer and analyze how the differences before Xoo inoculation contributes immune responses. In addition, the authors need to clarify how changes in miRNA abundance after Xoo infection are related to the difference in resistance.
Reply: Thanks for your positive comment, and we have revised it and added the special instructions in the updated manuscript (please see in line 168-171, line 180-181 and line 193-198).
2.Second, are time points indeed sufficient to characterize microRNA dynamics in the response to Xoo infection? In Fig. 1c, OsPBZ1 expression is obviously induced 4h after inoculation, then its expression is reduced 8h after inoculation. This result suggests that transcriptome changes are so dynamic from 4h to 8h after infection. Is it not necessary to test miRNA expressions during that time?
Reply: Thanks for your positive comment. In this study, the ideal time points of sampling were designed to reflect the most possible difference in transcription events in both rice genotypes under Xoo attacks, and only OsPBZ1 and OsWRKY45 out of four defense marked genes have been reported to involve in immune responses to Xoo infection (Huang et al., 2016;Tao et al., 2009). However, the postinvasive expression patterns of OsPBZ1 exhibited similar trends, except for 8 hpi, in both IR24 and IRBB5, and the lowest expression level of OsWRKY45-1, a representative negative regulator, occurred in 24 hpi after Xoo inoculation. Therefore, it is reasonable to choose 8 hpi and 24 hpi as best time points for investigating the miRNAs expression. (please see in line 347-349, line 352-353).
In addition to the above comments, I would like to raise several points that the authors should revise in order to make the arguments in the manuscript clearer.
3. (1) Line 125-127, the data provided in this paragraph are the result of small RNA sequencing. The result do not have any relationship with defense response. (2) Line 151-154, I can not agree that these results were confirmed to ….(3) In Figure 3B-D, it is better to compare between up-regulated miRNAs and to compare down-regulated miRNAs, separately (4) Line 185-191. miR167a-5p and miR167d-5p do not regulate common targets? This point is not clear in the manuscript. (5) In addition, the phrases “positive regulators” and “negative regulators” are used for describing miR167a-5p as a positive and miR167d-5p as a negative. These phrases confuse me. Please reconsider these phrases. (6) Samples used in figure 4A are not described. (7) Figure 4B must use colors with careful consideration of color weakness. (8) Figure 4C-H, a statistically significant deference compare to what? (9) Figure 5 seems not to easy to understand the authors argument. Rewrite the figure.
Reply:
(1) Thanks for your positive comment. One of the main goals for this study is to pinpoint whether small RNAs participate in rice-Xoo interaction, and the descriptive results of small RNA sequencing in line 125-l27 were used to expound that numerous small RNAs were involved in rice responses to Xoo.
(2) Thanks for your positive comment. We have reanalyze the data and revised our conclusion in the updated manuscript (please see line 155-158).
(3) Thanks for your positive comment, and we have revised Figure 3B-D and corresponding number of miRNAs in the updated manuscript (please see line 166-167, line 174-179, Figure 3B-G).
(4) Thanks for your positive comment, and we have add the relevant information in the updated manuscript (please see line 271-272, Figure 6E, 6F).
(5) Thanks for your positive comment, and we defined the positive regulators and negative regulators on the basis of the expression change character of miRNAs during Xoo infection. Even though osa-miR167a-5p and osa-miR167d-5p could cleave the transcripts of OsWD40-174, target gene LOC_Os04g57610 (OsARF8) may eventually determine the functional fate of osa-miR167a-5p after Xoo inoculation.
(6) Thanks for your positive comment, and we have described samples used in figure 4A in the updated manuscript (please see line 184-185, line 187).
(7) Thanks for your positive comment, and we have revised the colors of Figure 4B and Figure S2 by using yellow and blue (please see Figure 4B and Figure S2).
(8) Thanks for your positive comment, and we have added the control check information in the updated manuscript (please see line 190-192, line 221).
(9) Thanks for your positive comment, and we have rewritten the Figure 5, and added the special gene name instead of target genes families in the updated manuscript (please see line 281-282, Figure 5).
Round 2
Reviewer 2 Report
The revised manuscript seem to answer my requests raised to the previous manuscript.
I found one more question in the current manuscript.
In Figure 6 E and F, sequences osa-miR167a-5p and osa-miR167d-5p are completely same. Is it correct?
And I do not understand why osa-miR167d-5p can not cleave Os04g57610 while osa-miR167a-5p do.
In Figure 6, I suggest the authors to describe nucleotide sequences as RNA. Please use U instead of T.
In Figure 6 E, locus name in cleavage site position has typo, Os03g57610.
Author Response
I found one more question in the current manuscript.
In Figure 6 E and F, sequences osa-miR167a-5p and osa-miR167d-5p are completely same. Is it correct?
Reply: Thanks for your positive comment. Osa-miR167a-5p and osa-miR167d-5p have almost same sequences, except in the 21st position, and we have revised it in the updated manuscript (please see Figure 6E-F).
And I do not understand why osa-miR167d-5p can not cleave Os04g57610 while osa-miR167a-5p do.
Reply: Thanks for your positive comment. We have reanalyzed the degradome data, and found that the results from the manuscript originated from the alignment scores using Targetfinder software. According to the upper threshold of alignment scores ≦ 4, osa-miR167a-5p/LOC_Os04g57610 has got 4 points, and osa-miR167d-5p/LOC_Os04g57610 exceeded the ceiling due to the unmatched nucleotide base in 21st position, therefore osa-miR167a-5p could cleave Os04g57610 while osa-miR167d-5p did not.
In Figure 6, I suggest the authors to describe nucleotide sequences as RNA. Please use U instead of T.
Reply: Thanks for your positive comment, and we have revised it in the updated manuscript (please see Figure 6A-F).
In Figure 6 E, locus name in cleavage site position has typo, Os03g57610.
Reply: Thanks for your positive comment, and we have revised it in the updated manuscript (please see Figure 6D).